# Absorbing Material of Button Antenna with Directional Radiation of High Gain for P2V Communication

**DOI:** 10.3390/s23115195

**Published:** 2023-05-30

**Authors:** Yuanxu Fu, Tao Shen, Jiangling Dou, Zhe Chen

**Affiliations:** Faculty of Information Engineering and Automation, Kunming University of Science and Technology, Kunming 650032, China; shentao@kust.edu.cn (T.S.); chenzhe@kust.edu.cn (Z.C.)

**Keywords:** NZIM, ME structure, flat gain, quality factor Q

## Abstract

Vehicular communication systems can be used to enhance the safety level of road users by exchanging safety/warning messages. In this paper, an absorbing material on a button antenna is proposed for pedestrian-to-vehicle (P2V) communication, which provides safety service to road workers on the highway or in a road environment. The button antenna is small in size and is easy to carry for carriers. This antenna is fabricated and tested in an anechoic chamber; it can achieve a maximum gain of 5.5 dBi and an absorption of 92% at 7.6 GHz. The maximum distance of measurement between the absorbing material of the button antenna and the test antenna is less than 150 m. The advantage of the button antenna is that the absorption surface is used in the radiation layer of the antenna so that the antenna can improve the radiation direction and gain. The absorption unit size is 15 × 15 × 5 mm^3^.

## 1. Introduction

The cooperative intelligent transport systems (C-ITS) pilot deployment project, which has the objective of developing and experimenting with innovative road CITS solutions, offers pedestrian-to-vehicle (P2V) communication systems for vulnerable road user (VRU) protection [1,2,3,4,5,6,7,8]. To provide a good communication service to the P2V communication system, one should ensure that a minimum broadcast distance is achieved such that an adequate road safety level can be provided [9,10,11,12,13,14,15]. Therefore, directive antennas have been proposed to improve connectivity and security for vehicular communication and autonomous systems [16,17,18]. However, making the antenna smaller makes it easier to wear or carry. The button antenna is the most comfortable technology because it can be easily integrated into human clothes and provides a stable performance due to its rigidity [19,20]. Button antennas are used in many fields, such as in Internet of Things (IOT), medicine, Global Positioning System (GPS), and wireless body area network (WBAN) applications [21,22,23]. Researchers have focused on developing many types of button antennas. In [24], a wearable dual-port button antenna that excites pattern-diversity dual-polarized waves was proposed for ultra-wideband (UWB) applications. In [25], shorting vias are presented as standard components in planar technologies for textile button antennas. The high-profile button antenna was demonstrated for body-centric communication in [26], which resulted in a low gain value when tested on the body. A reconfigurable snap-on button antenna module with dedicated circuitry for wearable applications has been illustrated in [27]. This button antenna is an off-body antenna, starting with the radiating part, which is a monopole fed by a microstrip line for investigating the best structure based on a dielectric resonator [28]. In [29], a novel circularly polarized button antenna was proposed, covering the unlicensed national information infrastructure (U-NII) worldwide band. To date in the discussed button antennas, the antennas have been designed with large dimensions and less gain.

In this paper, an absorbing material for a button antenna is proposed. It mainly solves two problems of an antenna: antenna miniaturization and high gain directional radiation. The absorption rate of the button antenna can reach 92% at 7.6 GHz. It has an overall size of 15 × 15 × 5 mm^3^ and a measured gain of 5.5 dBi. This absorbing button antenna is suitable for short-distance communication. The structure of this paper is as follows. In the next section, the overall structure of the button antenna and its material composition are introduced. In Section 3, the absorption principle of near-zero-index metamaterials (NZIMs) is analyzed. To analyze the characteristics of the structure, Section 4 improves the radiation direction of the button antenna with metamaterials, and Section 5 analyzes the gain of the button antenna. Next, the SAR, S-parameters, efficiency, and gain of the measured and simulated results are assessed. Finally, the concluding remarks from this work are presented in Section 6.

## 2. Configuration

As shown in Figure 1, the overall size of the antenna is 15 × 15 × 5 mm^3^, where the grounding and feed isolation aperture is 2 mm and the coaxial feed diameter of the antenna is 1 mm. The antenna is composed of two substrates of glass fiber epoxy (FR-4) material. The FR-4 material has the advantages of a simple structure, easy fabrication, and good physical characteristics, meaning that the antenna radiation is not constrained by its own shape, allowing it to realize a stability of radiation. The permittivity of these substrates is 4.4, with a tangential loss of 0.027 and a thickness of 1.5 mm. A coaxial feed link is used to connect the parasitic patch and radiation patch to reduce coupling and impedance matching. The parasitic patch makes the antenna reach its maximum gain with a size of 13 × 13 mm^2^.

## 3. The Design and Analysis of Absorption Surface of the Button Antenna

This section mainly designs and analyzes the absorption material properties of the button antenna. Its main innovation comes from [30,31]. First, the near-zero-index metamaterial (NZIM) characteristics of a copper ring without embedded microstrip lines (CRO) and a copper ring with embedded microstrip lines (CRW) are analyzed. Second, the absorption rate and absorption angle of a copper ring with embedded microstrip lines (CRW) are analyzed.

Firstly, Figure 2a,c shows a copper ring without embedded microstrip lines (CRO) and a copper ring with embedded microstrip lines (CRW). The above two rectangular copper rings are symmetrical in structure, and the gap and microstrip line width are the same. Table 1 shows the parameter sizes of the two copper rings. Figure 2b,d shows the real and imaginary parts of the permittivity (ε), magnetic permeability (μ), and refractive index (*n*), which are approximately equal to zero by periodic electrical boundary (PEC) and periodic magnetic boundary (PMC) simulation results, which is obtained using Equations (Equation 1) and (Equation 2). The CRO and CRW belong to the NZIM. However, the coil without an embedded microstrip line (CRO) only belongs to the near-zero-index metamaterial at 7–7.6 GHz, while the inclusion of an embedded microstrip line (CRW) can improve the near-zero-index condition near 7.8 GHz.
(1)$$z=\pm \sqrt{\frac{{\left(1+S_{11}\right)}^{2}-S_{21}^{2}}{{\left(1-S_{11}\right)}^{2}-S_{21}^{2}}}$$
(2)$$n=\pm \frac{1}{kd}\left[\cos^{-1}\left(\frac{1-S_{11}^{2}+S_{21}^{2}}{2S_{21}}\right)\right]$$
where $S_{11}$ and $S_{21}$ are reflection and transmission coefficients. *n* is the refractive index, *k* is the wave number, d is the length of the unit cell, and z is the sign of effective impedance. the permittivity is ε = *n*/z, and the magnetic permeability is μ = n×z [32].

Secondly, Figure 3 shows the comparative absorption of CRO and CRW and the simulation model of CRW. The CRO and CRW is simulated by applying Floquet port and periodic boundary conditions in Ansys HFSS, and the absorptivity is obtained using Equation (Equation 3). In Figure 3b, the absorption of CRW is much better than that of CRO, and the absorption rate reaches 92% at 7.6 GHz. In the case of CRW, as the radiation surface of the button antenna has a certain slope, the slope absorption angle ((θ) and (ψ)) and absorption efficiency should be simulated in Figure 3a. Figure 4a,b shows that (θ) is the absorption rate after the change in the YZ slope angle, and (ψ) is the absorption rate after the change in the XY slope angle, which is the slope absorption rate of the CRW structure via PMC and PEC simulation. The slope angles vary from 0${}^{\circ }$ to 60${}^{\circ }$ with a step size angle of 15${}^{\circ }$. It can be observed that the proposed absorber material structure exhibits a different response at 7.6 GHz. The same absorptivity exists for different polarization angles (0${}^{\circ }$, 30${}^{\circ }$, 45${}^{\circ }$). However, when the incidence angle is 60${}^{\circ }$, the absorption rate begins to decrease, which is related to the resonant current at the resonant frequency. As shown in Figure 4c,d, the current density at an angle of 60${}^{\circ }$ significantly decreases, where red indicates the strong concentration and blue indicates the weak one. Because of the change in the absorption current of the ψ = 60${}^{\circ }$ angle, the absorption rate is in the complete absorption state.
(3)$$A\left(\omega \right)=1-|S_{11}{\left(\omega \right)|}^{2}-{\left|S_{21}\left(\omega \right)\right|}^{2}$$
where *A*(ω) is the absorptivity and $|S_{11}{\left(\omega \right)|}^{2}$ and $|S_{21}{\left(\omega \right)|}^{2}$ represent the reflectivity and transitivity, respectively, with respect to frequency ω [33].

## 4. The Radiation Direction of Button Antenna Is Improved by the CRO Metamaterial

In this section, as shown in Figure 5, one design uses a CRO metamaterial as the radiation layer of the antenna while the other is a common patch antenna. By comparing the radiation, directivity factor (D), and quality factor (Q) of the two antennas, the CRO metamaterial can improve the radiation direction. First, the equivalent circuit method is used to analyze the relationship between the quality factor (Q) and directivity factor (D). Second, the quality factor Q is used to analyze the CRO structure, and the index parameter that has the greatest influence on Q is found by analyzing the CRO structure. In addition, it provides a theoretical basis for improving the gain in the next section.

### 4.1. The Relationship between Q and D Is Analyzed According to the Equivalent Circuit Theory of Patch Antenna

In the equivalent circuit of the patch antenna, D is a directivity coefficient, which is defined as the ratio of the radiation intensity of the antenna in the maximum radiation direction to the average radiation intensity. The quality factor Q is a parameter that measures antenna bandwidth and gain. As shown in Equations (Equation 4) and (Equation 5), both are inversely proportional to the radiation conductance Gr; the other parameters are known numbers. Then, as shown in Figure 6, when the electromagnetic field model conforms to the ${\mathrm{TM}}_{01}$ mode and $2\lambda_{0}\geq a\geq 0.35\lambda_{0}$ (a is the length of the antenna and $\lambda_{0}$ is the free space wavelength), Q and D become positive, as shown in Equation (Equation 6). The results are shown in Figure 7a,b. After 7.6 GHz, the values of D and Q of the CRO antenna increase relative to the patch antenna, wherein the radiation direction of the co-polarization changes throughout Figure 7c,d. The change in the Q value is greater than that observed in D, so it is very important to study Q.
(4)$$D=\frac{2ab}{15\lambda_{0}G_{r}}$$
(5)$$Q=\frac{\varepsilon_{r}ab}{120\lambda_{0}hG_{r}}$$
where *a* and *b* are the length and width of the antenna patch, respectively. $G_{r}$ is the radiation conductance, $\varepsilon_{r}$ is the relative dielectric constant, $\lambda_{0}$ is the free space wavelength, and h is the antenna height [34].
(6)Q∝D

### 4.2. The Quality Factor Q of CRO Antenna Is Analyzed by an Equivalent Circuit

The change in the Q value can affect the gain of the whole antenna. Therefore, to improve the gain of the CRO antenna, we should first analyze the factors in the CRO structure that can affect the Q value, determine which factors have the greatest influence on Q value, and finally adopt the methods that further improve the Q value to increase the antenna gain. As shown in Figure 8, it is assumed that the copper coil in the CRO structure is the real part of the impedance and that the coupling between the coils is the imaginary part. Equation (Equation 7) is used to calculate the relationship between the Q, copper coil, and coupling distance. Therefore, $e_{i}$ is the coil width and $d_{i}$ is the coil coupling distance. Equation (Equation 8) is established based on the principle of the microstrip line between the width of copper coil and the real part. Equation (Equation 9) is established according to the parallel relation of the real part in Figure 8, and Equation (Equation 10) is substituted into Equations (Equation 8) to obtain Equation (Equation 9); in this way, the relationship between the coil width $e_{i}$ and the real part of impedance was obtained. When $e_{i}\geq h$ (h is the height of the coil), $Z_{real}$ is proportional to the sum of $e_{i}$ in the frequency range of 5–8 GHz, as shown in Figure 9a. Similarly, the relation between the coupling distance and imaginary part can be calculated using Equations (Equation 11) and (Equation 12). It can be concluded from Equation (Equation 12) that the imaginary part is proportional to the sum of the coupling distance $d_{i}$. The imaginary part and coupling distance shown in Figure 9b also conform to this relation. Substituting Equations (Equation 10) and (Equation 12) into Equation (Equation 7), we obtain Equation (Equation 13), which is the relationship between quality factor Q and the coupling distance $d_{i}$ and coil width $e_{i}$. The value of Q is proportional to the sum of the coil widths and is inversely proportional to the coupling distance. Therefore, increasing the coil width $e_{i}$ can further improve the Q value. As shown in Figure 10, the increase in the coil width $e_{i}$ is much greater than the decrease in the coupling distance $d_{i}$.
(7)$$Q=\frac{Z_{real}}{Z_{im}}$$

The $Z_{im}$ is the imaginary part of the impedance and $Z_{real}$ is the real part of the impedance [35].
(8)$$Z_{ri}{|}_{i=(1,2,3)}=\frac{120\pi }{e_{i}/h+2.42-0.44\times h/e_{i}+{\left(1-h/e_{i}\right)}^{6}}$$

The *h* is the height of the coil; we are assuming that it is a constant. $e_{i}$ (*i* = 1, 2, 3) is the width of coil [36].
(9)$$Z_{real}=\frac{1}{Z_{r1}}+\frac{1}{Z_{r2}}+\frac{1}{Z_{r3}}$$
(10)$$Z_{real}=\frac{\sum_{i=1}^{3}e_{i}/h+9.68-0.44\times \sum_{i=1}^{3}h/e_{i}+\sum_{i=1}^{3}{\left(1-h/e_{i}\right)}^{6}}{120\pi }$$
(11)$$Z_{mi}{|}_{i=(1,2,3)}=\frac{\varepsilon_{r1}\varepsilon_{r2}}{d_{i}}$$
where $\varepsilon_{r1}$ and $\varepsilon_{r2}$ is the dielectric constant of the two materials. $d_{i}$ (*i* = 1, 2, 3) is the coupling distance parameter of the coil, whose corresponding reactance is $C_{mi}$ (*i* = 1, 2, 3). The effect of the coil area on the capacitance is ignored [37].
(12)$$Z_{im}=\frac{1}{Z_{m1}}+\frac{1}{Z_{m2}}+\frac{1}{Z_{m3}}=\frac{\sum_{i=1}^{3}d_{i}}{\varepsilon_{r1}\varepsilon_{r2}}$$
(13)$$Q=\frac{\sum_{i=1}^{3}e_{i}}{\sum_{i=1}^{3}d_{i}}$$

## 5. Analysis of Gain of the CRW Structure of Button Antenna by Quality Factor Q

The main contents of this section are as follows: First, we analyze the improvement of the gain of the button antenna by quality factor Q. Second, the equivalent circuit method is used to analyze the embedded microstrip lines. Finally, the effect of its quality factor and CRW structure on the gain of the button antenna is analyzed. As seen in the previous section, the largest change in the CRO structure is the quality factor Q, as shown in Equation (Equation 14). In this way, the only parameter that affects antenna gain is the value Q. Therefore, it becomes an inevitable scheme to further improve the quality factor by embedding microstrip lines to improve the width of the copper coil of the CRO. The next step is to analyze the relationship between the $T_{1}$ length and quality factor Q and further analyze the change of the Q value after adding other embedded microstrip lines. According to Equation (Equation 7), Q is related to the real and imaginary parts of the impedance. Therefore, this chapter mainly analyzes the impedance after adding the $T_{1}$ symmetrical microstrip line. The first part is the real part analysis, and the second part is the imaginary part analysis; finally, all microstrip lines ($T_{2}$, $T_{3}$, $P_{1}$, $P_{2}$, $P_{3}$) are added to improve the antenna gain.
(14)$$G=\frac{DQ}{Q_{r}}$$
where D is the antenna orientation coefficient; the D is constant is 6 when the antenna size is less than 0.35$\lambda_{0}$ [38]. *Q* is the quality factor and $Q_{r}$ is the radiation loss ($Q_{r}$ is also a constant).

### 5.1. Analysis of Real and Imaginary Parts of Impedance with the Addition of T_*1*_

As shown in Figure 11, the real part of the microstrip line is analyzed according to the principle of the equivalent circuit. According to the equivalent hypothesis in [39], the whole circuit is divided into the T-type circuit and π-type circuit so that the whole circuit is assumed to be a symmetrical homogeneity circuit; the change in the whole circuit after adding $T_{1}$ means adding impedance $Z_{T2}$. The process is as follows:(1)The relationship between $T_{1}$ and the real part of the impedance of T-type and π-type is calculated using Equations (Equation 15)–(Equation 17);(2)The impedance Equation (Equation 18) of the real part is listed by the circuit diagram in Figure 11;(3)Taking the derivative of Equation (Equation 18), Equations (Equation 19) and (Equation 20) are obtained, from which it can be seen that the impedances $Z_{T1}$ and $Z_{T2}$ are directly proportional to the real part of the overall impedance, meaning that the length of $T_{1}$ is directly proportional to the overall impedance.

After calculation, the following conclusion can be drawn: $T_{1}$ is inversely proportional to the real part of the impedance from 2 to 5.5 GHz in Figure 12a, which belongs to the π-type circuit; and $T_{1}$ is directly proportional to the real part of the impedance from 6.5 to 8.5 GHz in Figure 12b, which belongs to the T-type circuit. In the same way, the relation between the imaginary part of the impedance and the length of $T_{1}$ was calculated using Equations (Equation 21)–(Equation 23) and is shown in Figure 13. The conclusions are as follows: $T_{1}$ is proportional to the imaginary part impedance of the T-type and inversely proportional to the imaginary part impedance of the π-type. As shown in Figure 14, when $T_{1}$ continuously increases, the imaginary part of the impedance frequency from 2 to 5.5 GHz gradually decreases. Therefore, it is determined that this property belongs to a π-type circuit. In the same way, it belongs to a T-type circuit at 6–8.5 GHz. As shown in Figure 15, after adding $T_{1}$, Q is not improved at 2–5 GHz, whereas Q is greatly improved at 6–8 GHz. Therefore, the parameters of $T_{2}$, $T_{3}$, $P_{1}$, $P_{2}$, and $P_{3}$ are optimized based on a T-type circuit.
(15)$$\alpha =\beta T_{1}$$
where α is the phase, β is the attenuation coefficient of microstrip line, and $T_{1}$ is the load microstrip line length [40,41].
(16)$$\left\{\begin{array}{cc} Z_{T1}=Z_{0}tan\left(\alpha /2\right) & Z_{T2}=Z_{0}\left[1/sin\left(\alpha \right)\right]\;\;\;T-\mathrm{type}\;\mathrm{circuit}\\ Y_{\pi 1}=Y_{0}tan\left(\alpha /2\right) & Y_{\pi 2}=Y_{0}\left[1/sin\left(\alpha \right)\right]\;\;\;\pi -\mathrm{type}\;\mathrm{circuit} \end{array}\right.$$

The $Z_{0}$ is the microstrip line impedance constant, and $Y_{0}$ is the microstrip line admittance constant [42,43].
(17)$$\left\{\begin{array}{cc} Z_{T1}=Z_{0}tan\left[\left(\beta T_{1}\right)/2\right] & Z_{T2}=Z_{0}\left[1/sin\left(\beta T_{1}\right)\right]\\ Y_{\pi 1}=Y_{0}tan\left[\left(\beta T_{1}\right)/2\right] & Y_{\pi 2}=Y_{0}\left[1/sin\left(\beta T_{1}\right)\right] \end{array}\right.$$
(18)$$\left\{\begin{array}{c} Z_{Treal}=\left(2Z_{T1}^{2}Z_{T2}+3Z_{T2}^{2}Z_{T1}\right)/\left(2Z_{T1}^{2}+4Z_{T1}+Z_{T2}^{2}\right)+Z_{T1}\\ Y_{\pi real}=\left(Y_{\pi 1}^{3}+4Y_{\pi 1}^{2}Y_{\pi 2}+2Y_{\pi 1}Y_{\pi 2}^{2}\right)/\left(4Y_{\pi }^{4}+6Y_{\pi 1}Y_{\pi 2}+2Y_{\pi 2}^{2}\right) \end{array}\right.$$
(19)$$\left\{\begin{array}{c} \partial Z_{Treal}/\partial Z_{T1}=2Z_{T1}^{2}Z_{T2}^{2}+3Z_{T2}^{2}+4Z_{T1}Z_{T2}^{2}\\ \partial Z_{Treal}/\partial Z_{T2}=4Z_{T1}^{4}+10Z_{T2}^{2}Z_{T1}^{2}+12Z_{T1}^{3}Z_{T2} \end{array}\right.$$
(20)$$\left\{\begin{array}{c} \partial Y_{\pi real}/\partial Y_{\pi 1}=Y_{\pi 1}^{4}+12Y_{\pi 1}^{3}Z_{\pi 2}+36Y_{\pi 1}^{2}Y_{\pi 2}^{2}+44Y_{\pi 1}Y_{\pi 2}^{3}+8Y_{\pi 2}^{4}\\ \partial Y_{\pi real}/\partial Y_{\pi 2}=10Y_{\pi 1}^{4}+14Y_{\pi 1}^{2}Y_{\pi 2}^{2}+12Y_{\pi 1}^{3}Y_{\pi 2} \end{array}\right.$$
(21)$$\left\{\begin{array}{ccc} X_{L/2}=Z_{0}tan\left(\alpha /2\right) & C_{i}=sin\left(\alpha \right)/Z_{0} & T-\mathrm{type}\;\mathrm{circuit}\\ X_{L}=Z_{0}sin\left(\alpha \right) & C_{i/2}=tan\left(\alpha /2\right)/Z_{0} & \pi -\mathrm{type}\;\mathrm{circuit} \end{array}\right.$$

$X_{L}$ is inductance of the imaginary part impedance, and $C_{i}$ is the capacitance of the imaginary part impedance [44,45].
(22)$$Z_{im}=\sqrt{\frac{X_{L}}{C_{i}}}$$
(23)$$\left\{\begin{array}{c} Z_{imT}=Z_{0}/cos\left(\beta T_{1}/2\right)\\ Z_{im\pi }=Z_{0}\cdot cos\left(\beta T_{1}/2\right) \end{array}\right.$$

### 5.2. Final Result Analysis

To further improve the antenna gain, microstrip lines are continuously embedded. As shown in Figure 16, case 1, case 2, and case 3 show the current distribution. With the addition of embedded microstrip lines, the current density and intensity increase leads to an increase in antenna gain. As shown in Figure 16a, with the addition of microstrip lines $T_{1}$, $T_{2}$, $T_{3}$, $P_{1}$, $P_{2}$, and $P_{3}$, the value of 1/Q is progressively decreasing; therefore, we have to optimize for the smallest 1/Q, which is the maximum value of Q. HFSS software was adopted for the optimization, and the minimum value of 1/Q at 7.6 GHz is shown in Figure 16b. Figure 16c verifies that the maximum value of Q is the maximum value of the antenna gain by gain of the co-planar polarization.

## 6. Measured and Simulated Results

In this section, the performance of the button antenna is evaluated on the human body. The absorptivity, S-parameters, efficiency, and radiation pattern are analyzed together with the SAR values. The scenario of the numerical model is shown in Figure 17. The cubic tissue model consists of a 1.7 mm thick skin layer, a 5 mm thick fat layer, a 20 mm thick muscle layer, and a 13 mm thick bone layer with a size of 150 × 150 mm^2^. All of the material parameters of the tissue were obtained from the values available in the CST Studio human tissues library. A 5 mm air gap was set between the antenna and the surface of the skin to mimic any worn clothes. An R&S microwave signal generator was used to test the $S_{11}$ parameters. The gain was measured using the antenna anechoic chamber with an signal generator R&S microwave signal generator, which has a maximum output power of 11 dBm. Figure 18 shows the physical antenna and measurement environment.

### 6.1. Specific Absorption Rate (SAR) and Absorptivity Property Analysis

For wearable applications, the SAR value is a critical parameter. The numerical SAR distributions of the proposed antennas are shown in Figure 19. The calculation is based on the IEEE C 95.3 standards, and the simulated power is 200 mW. According to the European standard, the SAR results should be under 2 W/kg averaged over 10 g of tissue. As shown in Figure 19, (a) the SAR peak value is 0.18 W/kg at 7.6 GHz and (b) the SAR peak value is 0.10 W/kg at 7.8 GHz, both of which meet the standard. The absorption efficiency is calculated by measuring the S-parameter of the antenna radiation surface. The absorption efficiency measurement is basically consistent with the simulation in Figure 20. In Table 2, compared with other button antennas, this button antenna has a smaller SAR and has less influence on body tissues.

### 6.2. Analysis of S-Parameters, Efficiency, and Radiation Pattern on the Body, Free Space and Textile

As shown in Figure 21, (a) is the S-parameter value of the button antenna in different wearing substances, such as free space, arms of the human body, and cloth textiles. Judging from the S-parameter, this does not have much influence on the body tissue and clothing textile. The effect of the antenna radiation on the free space, human body, and cloth textiles is shown in Figure 21b–d. Although the human body and clothes improve antenna radiation and gain, the increase is 0.1 dBi. That is, it has no effect on the antenna radiation itself. As shown in Figure 22, other peak gains and the efficiency are unchanged, so wearing the button antenna with different substances has no influence on the performance and absorption. Compared with other button antennas, this button antenna has a small size, high gain, and good efficiency, as shown in Table 2. In other words, the antenna provides a good stable performance.

## 7. Conclusions

An absorbing material on a button antenna for P2V communication was proposed, fabricated, and measured, which provides safety service to road workers on highways or in a road environment. The maximum distance of measurement between the absorbing material of the button antenna and the test antenna is less than 150 m, which is a safe distance for traffic. This antenna can achieve a maximum gain of 5.5 dBi and absorption of 92% at 7.6 GHz. Compared with other similar antennas in Table 2, the proposed antenna is very compact in size and achieves an excellent gain and reasonable efficiency; the bandwidth is only 4%. The modified absorbing material structure and detail considerations for wearable scenarios make it a very suitable candidate for on-body applications.

## Figures and Tables

**Figure 1 sensors-23-05195-f001:**
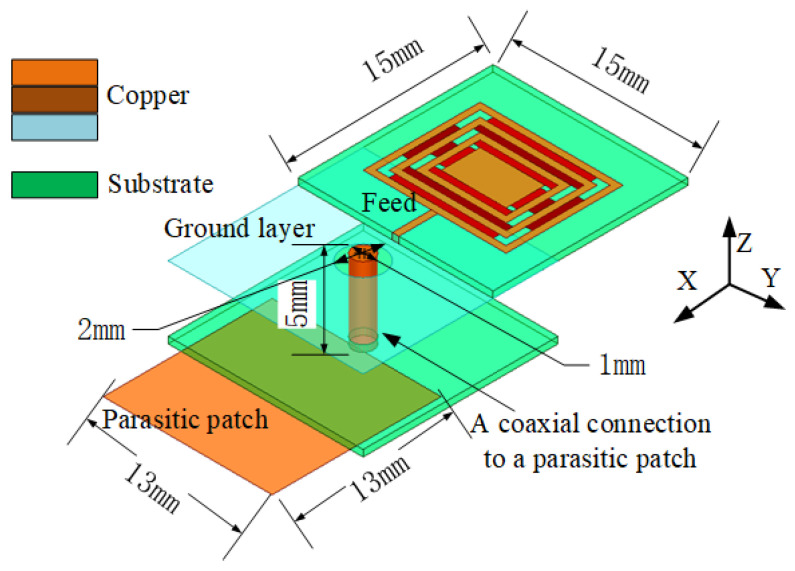
Button antenna using absorbing material configuration.

**Figure 2 sensors-23-05195-f002:**
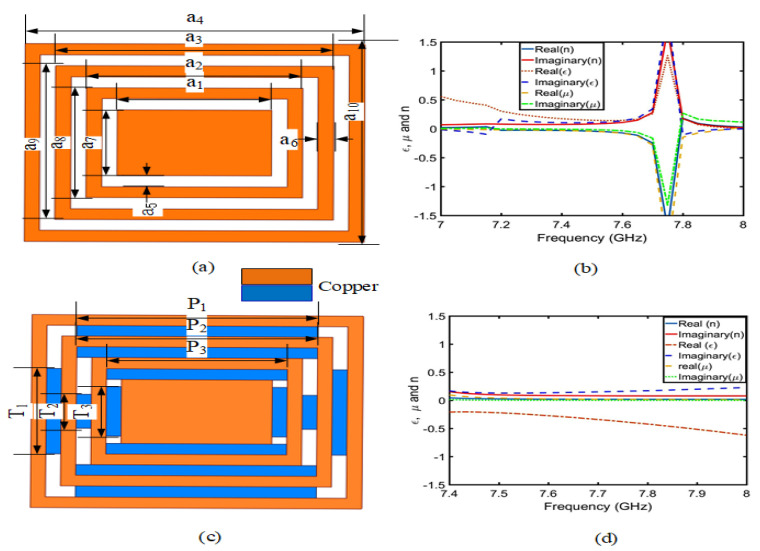
CRO and CRW models of NZIM: (**a**) CRO model, (**b**) CRO of ε, μ, and *n* parameters simulation, (**c**) CRW model, and (**d**) CRW of the ε, μ, and *n* parameters simulation.

**Figure 3 sensors-23-05195-f003:**
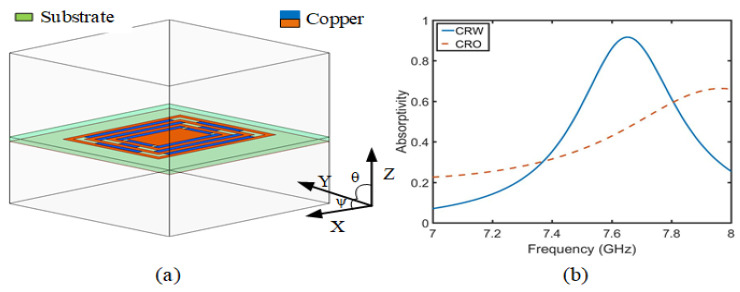
(**a**) Simulation model of CRW absorption and (**b**) the absorption rate of CRW and CRO.

**Figure 4 sensors-23-05195-f004:**
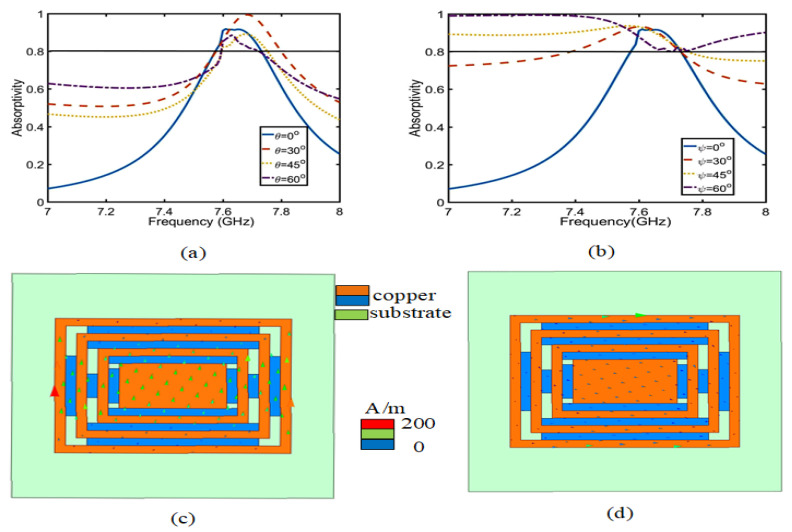
(**a**,**b**) are slope absorption rate of θ and ψ. (**c**,**d**) are the current distribution of θ = 60${}^{\circ }$ and ψ = 60${}^{\circ }$, respectively.

**Figure 5 sensors-23-05195-f005:**
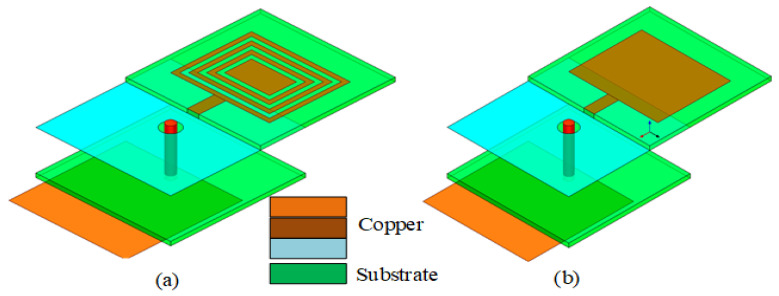
(**a**) CRO metamaterial antenna; (**b**) common patch antenna.

**Figure 6 sensors-23-05195-f006:**
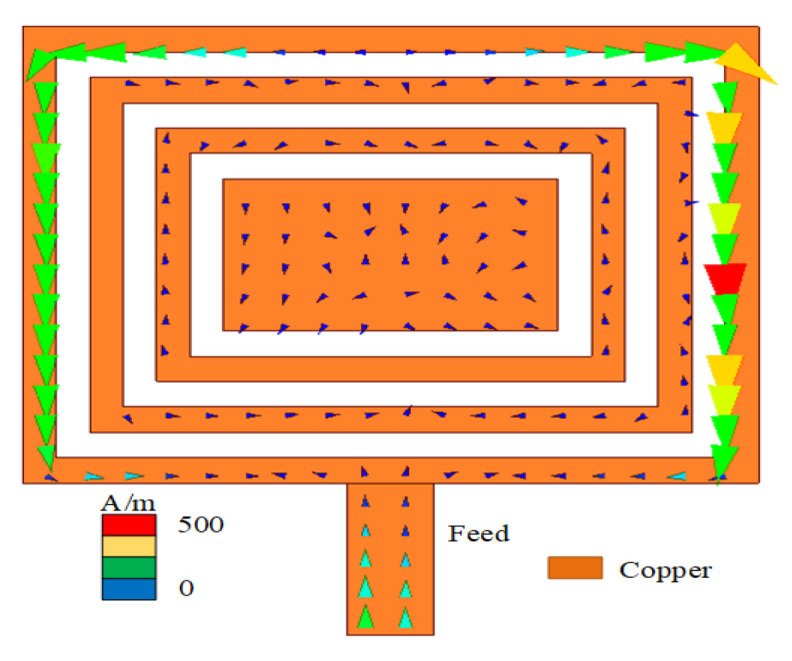
The surface current of the CRO metamaterial.

**Figure 7 sensors-23-05195-f007:**
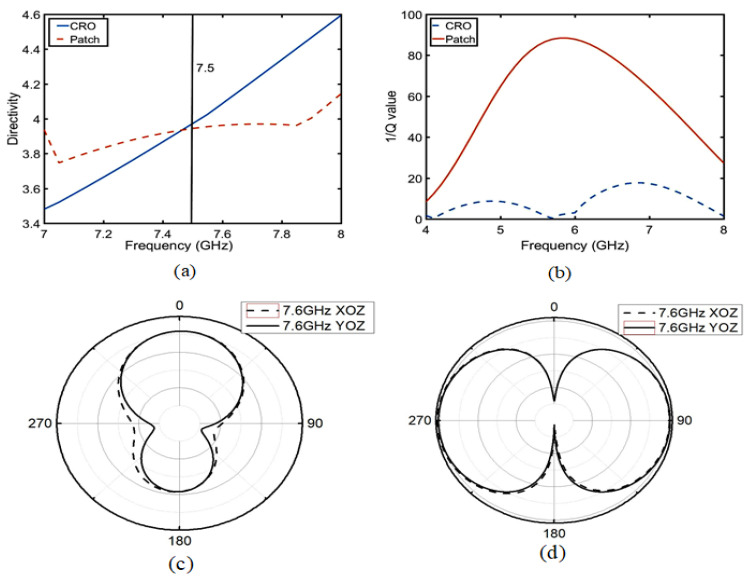
Comparison of antenna parameters of the CRO atennna and patch antenna: (**a**) is the directivity factor and (**b**) is the quality factor Q. At 7.6 GHz, (**c**) is the directional radiation pattern of the CRO metamaterial and (**d**) is the omnidirectional radiation pattern of the patch antenna.

**Figure 8 sensors-23-05195-f008:**
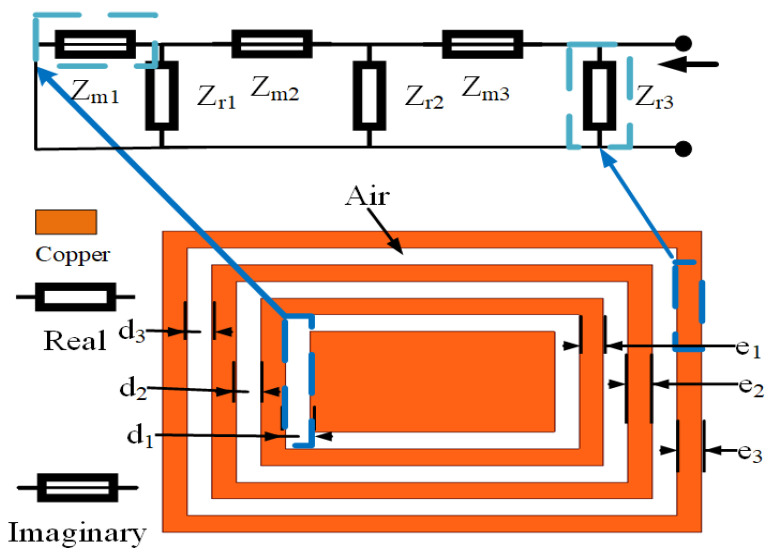
Equivalent circuit diagram of the CRO structure.

**Figure 9 sensors-23-05195-f009:**
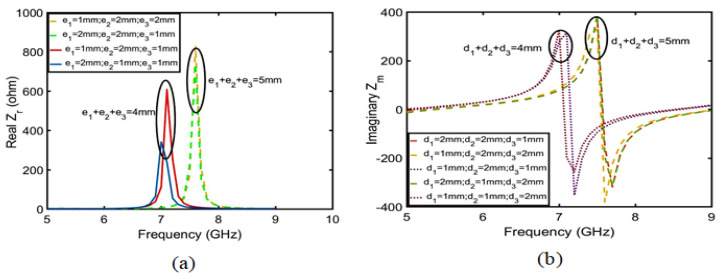
(**a**) is the relationship between the sum of $e_{i}$ and the real part. (**b**) is the relationship between the sum of $e_{i}$ and the imaginary part.

**Figure 10 sensors-23-05195-f010:**
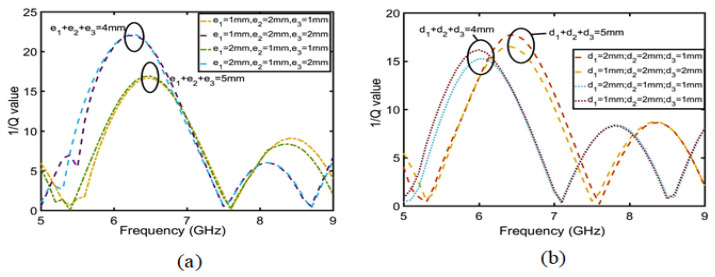
(**a**) is the relationship between the sum of $e_{i}$ and 1/Q. (**b**) is the relationship between the sum of $d_{i}$ and 1/Q.

**Figure 11 sensors-23-05195-f011:**
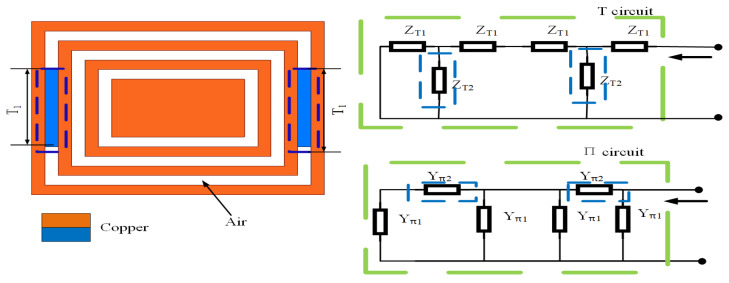
Real part analysis diagram of the rectangular metal ring structure after embedding T1 of the CRO.

**Figure 12 sensors-23-05195-f012:**
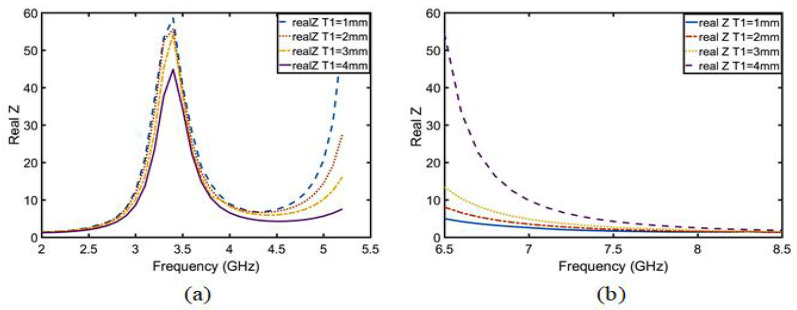
Real part analysis diagram of impedance after embedding $T_{1}$ at (**a**) 2–5.5 GHz and (**b**) 6.5–8.5 GHz.

**Figure 13 sensors-23-05195-f013:**
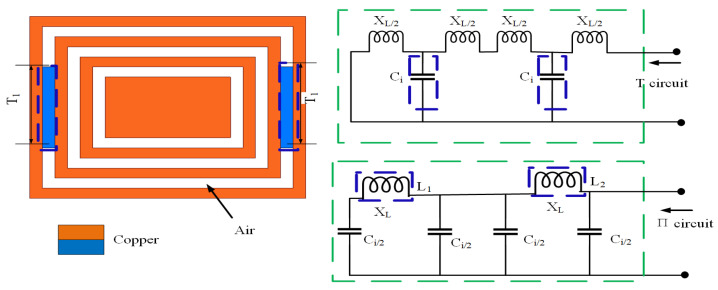
Imaginary part analysis diagram of the rectangular metal ring structure after embedding $T_{1}$ of the CRO.

**Figure 14 sensors-23-05195-f014:**
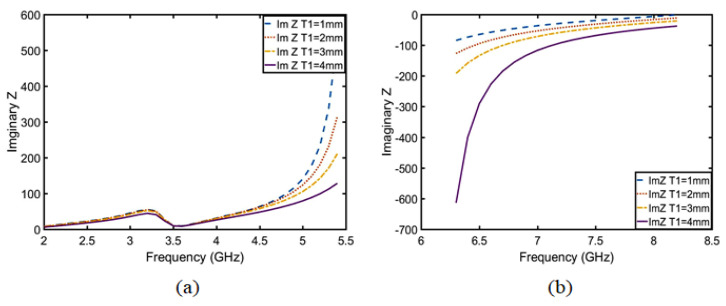
Imaginary part analysis diagram of impedance after embedding $T_{1}$ at (**a**) 2–5.5 GHz and (**b**) 6.5–8.5 GHz.

**Figure 15 sensors-23-05195-f015:**
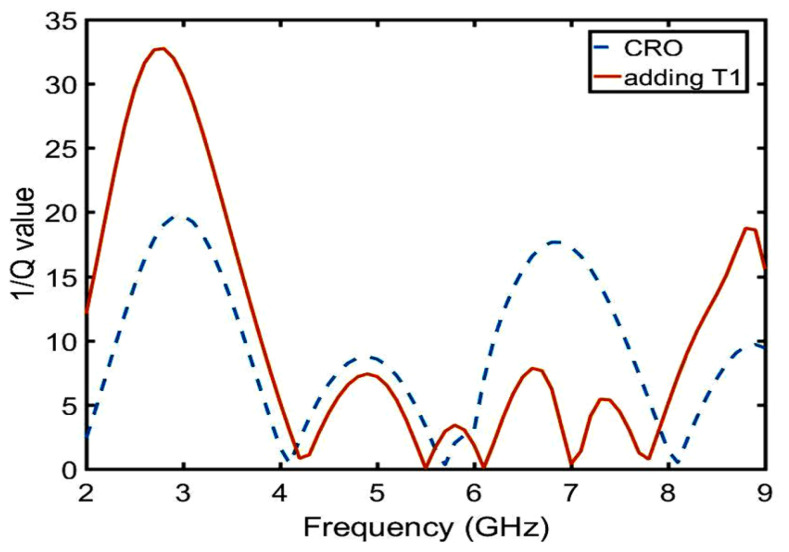
Comparison of 1/Q values between embedding the $T_{1}$ of the CRO and CRO.

**Figure 16 sensors-23-05195-f016:**
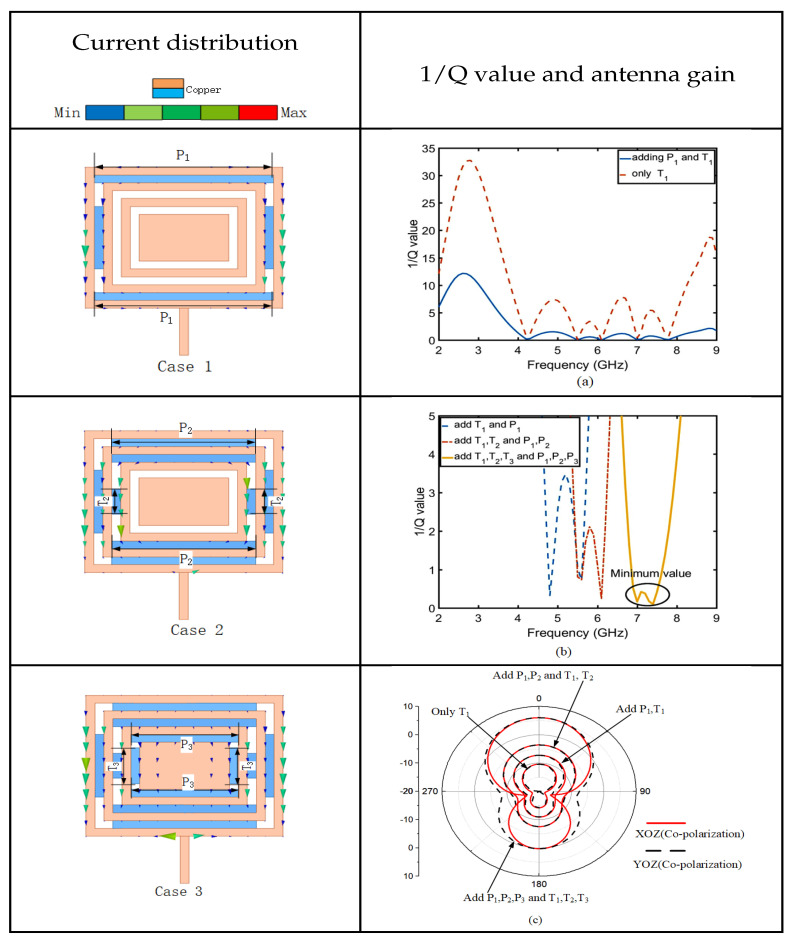
CRW analysis: adding the $P_{1}$ current distribution (case 1); adding the $P_{2}$ and $T_{2}$ current distribution (case 2); adding the $P_{3}$ and $T_{3}$ current distribution (case 3); comparison diagram of 1/Q after adding $P_{1}$ (**a**); the minimum value of 1/Q after CRW optimization (**b**); maximum gain after CRW optimization (**c**).

**Figure 17 sensors-23-05195-f017:**
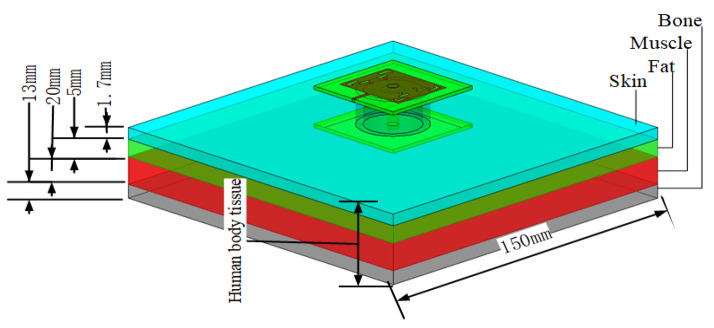
Four-layered human tissue.

**Figure 18 sensors-23-05195-f018:**
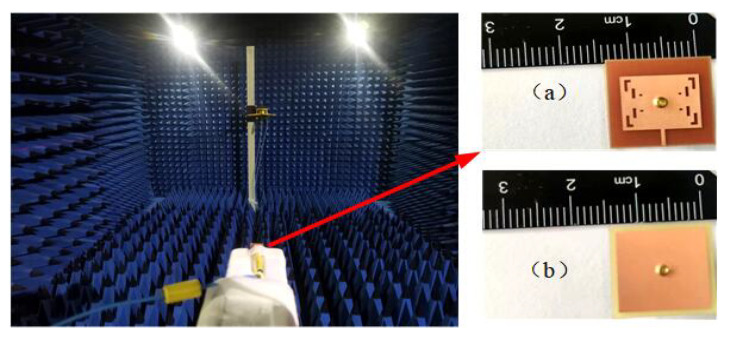
The (**a**) front and (**b**) back of the antenna object in test environment.

**Figure 19 sensors-23-05195-f019:**
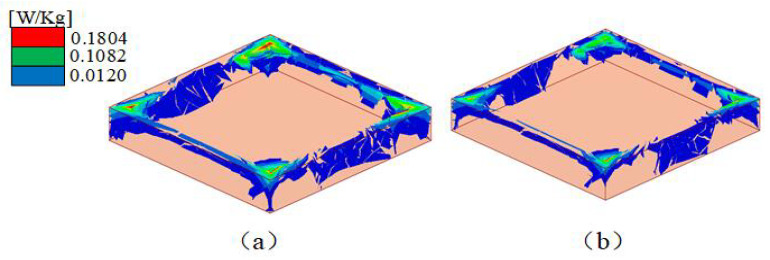
Simulated SAR of the button antenna on the human body at (**a**) 7.6 GHz and (**b**) 7.8 GHz.

**Figure 20 sensors-23-05195-f020:**
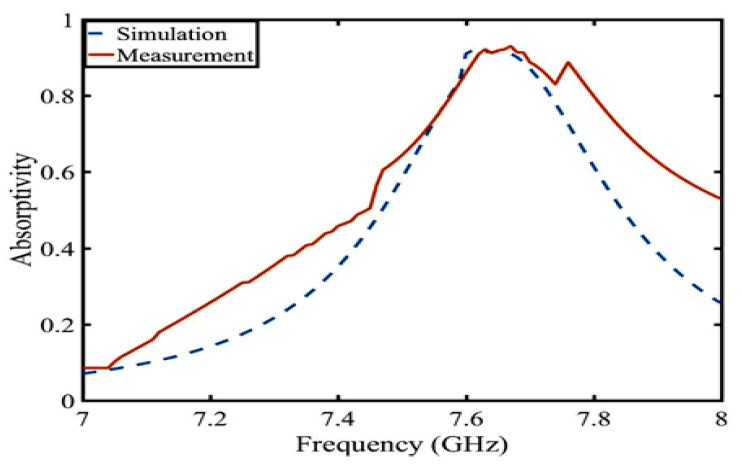
Simulation and measurement of absorptivity.

**Figure 21 sensors-23-05195-f021:**
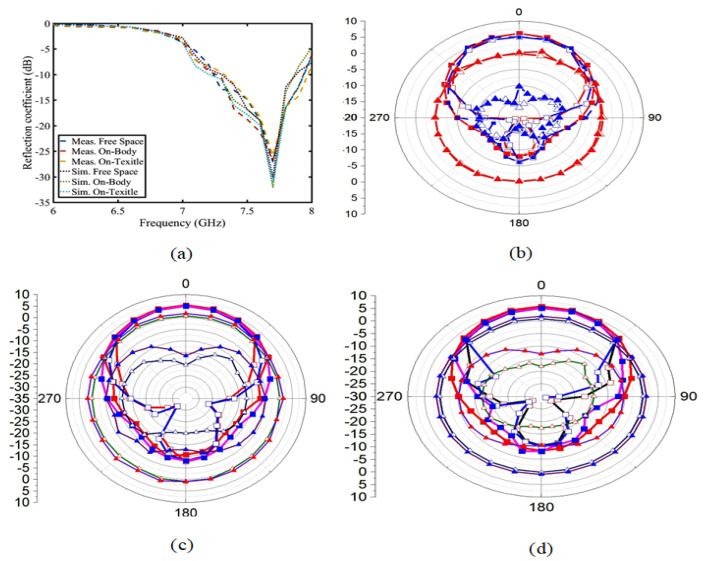
(**a**) is the $S_{11}$ simulation and measurement diagram. (**b**–**d**) are the radiation simulation and measurement of the button antenna on the free space, on-body, and on-textile regions. (□ is the co-polarization simulation XOZ, ▪ is the co-polarization measure XOZ, □ is the co-polarization simulation YOZ, ▪ is the co-polarization measure YOZ, ▵ is the cross-polarization simulation XOZ, ▴ is the cross-polarization measure XOZ, ▵ is the cross-polarization simulation YOZ, and ▴ is the co-polarization measure YOZ).

**Figure 22 sensors-23-05195-f022:**
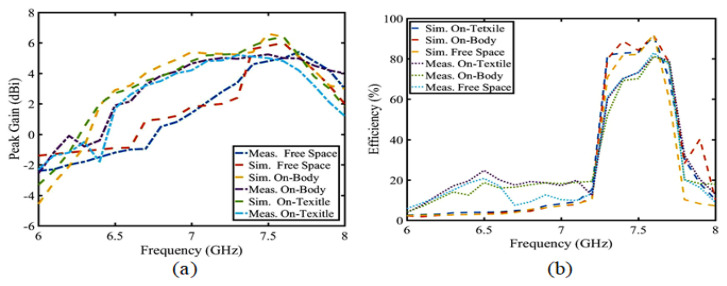
Simulation and measurement: (**a**) peak gain and (**b**) efficiency.

**Table 1 sensors-23-05195-t001:** Dimensions of the CRO and CRW.

Parameter	Value (mm)	Parameter	Value (mm)
$a_{1}$	5	$P_{1}$	8
$a_{2}$	7	$P_{2}$	8
$a_{3}$	9	$P_{3}$	6
$a_{4}$	11	$T_{1}$	2.3
$a_{5}$	5	$T_{2}$	1.6
$a_{6}$	5	$T_{3}$	4
$a_{7}$	3	$a_{9}$	7
$a_{8}$	5	$a_{10}$	9

**Table 2 sensors-23-05195-t002:** Performance comparison with the literature.

Reference	$\lambda_{0}$ × $\lambda_{0}$ × h mm^3^	BW (%)	Radiation Pattern	Gain (dBi)	SAR (W/kg)	Efficiency (%)
[20]	0.32 × 0.30 × 1	31	O	2.1	-	78
[21]	0.27 × 0.27	2.1	D	2.5	1.6	82
[22]	0.32 × 0.30 × 1.8	5	D	3.3	0.5	75
[23]	0.34 × 0.34 × 6.5	5	O	2	-	84
[24]	0.36 × 0.36 × 4.0	15	D	2	1.0	90
[25]	0.45 × 0.45 × 1.2	5.3	D	4.2	0.74	78
[26]	2.5 × 2.5 × 0.2	3	D	3.9	-	77
[27]	1.5 × 1.5 × 1.2	4	D	3.0	-	75
[28]	1.2 × 1.5 × 0.5	5	O	2.8	-	70
[29]	2.1 × 2.1 × 0.5	8	D	1.16	1.12	78
This work	0.24 × 0.24 × 5	4	D	5.5	0.18	92

O is the omnidirectional radiation and D is the directional radiation.

## Data Availability

Not applicable.

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
