# Peer review of "Absorbing Material of Button Antenna with Directional Radiation of High Gain for P2V Communication"

_sensors, 2023, doi:10.3390/s23115195_

Round 1

Reviewer 1 Report

·         I want to thank you for the work you have done to clarify the creation of Metamaterial Absorbing of Button Antenna with Directional Radiation of High Gain for Pedestrian to Vehicle Communication. You demonstrate your technical and scientific skills. It is decisive for me that you have relevant results in the given research. I want to draw your attention to the fact that the abstract in which you demonstrate the meaningfulness of scientific conclusions is very focused on determining your experiment and not clarifying your improvements and what is the goal of your manuscript Pedestrian to Vehicle Communication. This generally discourages the reader from continuing to read it.

·         Near-zero-index absorbing metamaterial of button antenna is proposed. It mainly solves two problems of the antenna, one is antenna miniaturization and other has directional radiation of high gain. This solution is useful, but it is just another wireless network solution. If the authors would at least point out in the article where such a construction is suitable practical application. It would be more exceptional for the reader than a lengthy description of the creation of such an antenna.

·         Does the FR-4 material have loss? Could you provide the radiation efficiency of your antenna at the operating frequency?

·         There is no clear explanation on Fig.1? What is the difference between the excitation feed and coaxial feed from center. Even in prototype it is not connected to any port extension

·         In the text some places used the word called formulae and other places equation. It should be maintained uniformly.

·         Fig.7(b) mentioned as omnidirectional pattern but it is not

·         The authors could explain the purpose of modernizing this type of antenna in the article. Who is it for? If someone decides on this type of antenna, in what environment and with what parameters can they be used.

·         State of the art is not sufficient. Add some additional relevant literature related to FSS and high gain/directivity antennas.

If the authors can revise Sections1, 2, and 6 in particular, it is possible to consider that the article will acquire an interesting character for the professional public in the field of antenna technology. Also, I would suggest you to modify the title as well.

Author Response

 1. I want to thank you for the work you have done to clarify the creation of Metamaterial Absorbing of Button Antenna with Directional Radiation of High Gain for Pedestrian to Vehicle Communication. You demonstrate your technical and scientific skills. It is decisive for me that you have relevant results in the given research. I want to draw your attention to the fact that the abstract in which you demonstrate the meaningfulness of scientific conclusions is very focused on determining your experiment and not clarifying your improvements and what is the goal of your manuscript Pedestrian to Vehicle Communication. This generally discourages the reader from continuing to read it.

R: my article is mainly button antenna, miniaturized antenna, so according to the requirements to miniaturized or button antenna based discussion. Here I have all modified in the Introduction and abstract as a basis for this article. Pedestrian and vehicle is only an application environment, so it is more about the experimental research of button antenna. This antenna is short-range communication, so it is more about the process of experiment than the change of environment.

 Modify the introduction and abstract as follows: line 1-43

Vehicular communication systems can be used to enhance the safety level of road users by exchanging safety/warning messages. In this paper, an absorbing metamaterial on a button antenna is proposed for pedestrian-to-vehicle (P2V) communication,providing safety service to road workers on the highway or in a road environment. This antenna is fabricated and tested in an anechoic chamber. It can achieve a maximum gain of 5.5 dBi and absorption of 92\% at 7.6 GHz. The maximum distance between the absorbing metamaterial of the button antenna and the test antenna is less than 150 m. The advantage of the button antenna is that the absorption surface is used in the radiation layer of the antenna, so that the antenna can improve the radiation direction and gain. The absorption unit size is 15*15*5 mm3.

Introduction

Therefore, directive antennas have been proposed to improve connectivity and security for vehicular communication and autonomous systems [14]-[16]. However, making the antenna smaller makes it easier to wear or carry. The button antenna is the most comfortable technology because it can be easily integrated into human clothes and provides stable performance due to its rigidity[17] [18]. Button antennas are used in many fields such as the internet of things (IOT), medicine, global positioning system (GPS) and wireless body area network (WBAN) applications [19]-[21].  Researchers have focused on developing many types of button antennas. In [21], a wearable dual-port button antenna that excites pattern-diversity dual-polarized waves is proposed for ultra-wideband (UWB) applications. In [22],  shorting vias are standard components in planar technologies for textile button antennas. The high-profile button antenna demonstrated for body-centric communication in [23]. This antenna resulted in a low gain value when tested on the body. A reconfigurable snap-on button antenna module with dedicated circuitry for wearable applications has been illustrated [24]. This button antenna is an off-body antenna, starting with the radiating part which is a monopole fed by a microstrip line for investigating the best structure based on a dielectric resonator [25]. In [26], a novel circularly polarized button antenna is proposed, covering the unlicensed national information infrastructure (U-NII) worldwide band. To date in the discussed button antennas, the antennas have been designed with large dimensions and less gain.

2. Near-zero-index absorbing metamaterial of button antenna is proposed. It mainly solves two problems of the antenna, one is antenna miniaturization and other has directional radiation of high gain. This solution is useful, but it is just another wireless network solution. If the authors would at least point out in the article where such a construction is suitable practical application. It would be more exceptional for the reader than a lengthy description of the creation of such an antenna.

R: As this antenna is a miniaturized antenna with limited radiation distance, most applications are based on short range communication. Therefore, it has been written in the abstract that its working distance is within 150 meters.

Abstract : line 6-7

The maximum distance between the absorbing metamaterial of the button antenna and the test antenna is less than 150 m

3.Does the FR-4 material have loss? Could you provide the radiation efficiency of your antenna at the operating frequency?

R: FR-4 is the loss of tangential Angle, which is the actual error loss in its own engineering quantification and simulation. It is proposed here that it is mainly used in simulation and measurement. Figure. 21 shows the radiation efficiency of the antenna and the final efficiency produced by different contact media.

Configure line 49-52

The permittivity of these substrates is 4.4, with a tangential loss of 0.0027 and a thickness of 1.5 mm. A coaxial feed link is used to connect the parasitic patch and the radiation patch to reduce coupling and impedance matching.

4. here is no clear explanation on Fig.1? What is the difference between the excitation feed and coaxial feed from center. Even in prototype it is not connected to any port extension.

R: The parasitic patch below the coaxial feed link is described here, which is used to regulate impedance matching. Here it is more clear to change the simulation diagram, and the processing diagram was used before. Produces certain visual errors.

Figure 1 has been modified.

Configure line: P 52-53.

 5.In the text some places used the word called formulae and other places equation. It should be maintained uniformly.

R: Thank you for reminding me. It has been revised comprehensively.

line: P 94-95,100-113,119,152-155.

6.Fig.7(b) mentioned as omnidirectional pattern but it is not

 R: This is a two-dimensional polarization diagram, it just shows the same polarization, it doesn't show the cross polarization, so it's an omnidirectional antenna. But what I'm saying here is that metamaterials have the problem of improving the direction of the co-polarized radiation, so I'm not saying cross polarization again. Figure 20 shows this phenomenon in detail, and shows the contrast between cross-polarization and co-polarization.

The radiation direction of button antenna is improved by CRO metamaterial. Line:98-100

The results are shown in Figure 7 (a) and (b). After 7.6 GHz, the values of D and Q of the CRO antenna increase relative to the patch antenna, in which the radiation direction of co-polarization changes through Figure 7 (c) and (d). The change in the Q value is greater than that in D, so it is very important to study Q.

7.The authors could explain the purpose of modernizing this type of antenna in the article. Who is it for? If someone decides on this type of antenna, in what environment and with what parameters can they be used.

R: Firstly, miniaturized antenna is a hot topic in current research, not only for individual, but also for its convenience and portability. Button antenna is mainly used in clothes and related woven articles. The test shows that the antenna has radiation stability and miniaturization characteristics. Secondly, at present, my research is only used in short distance communication. Here, it is used in the car identification system to test the identity of the owner and a test point for pedestrian safety.

Introduction.line: P 12-33

In this paper, an absorbing metamaterial for a button antenna is proposed. It mainly solves two problems of the antenna: antenna miniaturization and high gain directional radiation . The absorption rate of the button antenna can reach 92\% at 7.6 GHz. It has an overall size of 15$\times$15$\times$5 $\rm mm^3$ and a measured gain of 5.5 dBi.This absorbing button antenna is suitable for short distance communication.

8.State of the art is not sufficient. Add some additional relevant literature related to FSS and high gain/directivity antennas.

R: This modification is mainly based on button antenna. Some details of FSS metamaterials are not stated here. The whole work is more integrated and relative comparison, otherwise the experiment will be biased.

Introduction line: P 18-32

Button antennas are used in many fields such as the internet of things (IOT), medicine, global positioning system (GPS) and wireless body area network (WBAN) applications [19]-[21].  Researchers have focused on developing many types of button antennas. In [21], a wearable dual-port button antenna that excites pattern-diversity dual-polarized waves is proposed for ultra-wideband (UWB) applications. In [22],  shorting vias are standard components in planar technologies for textile button antennas. The high-profile button antenna demonstrated for body-centric communication in [23]. This antenna resulted in a low gain value when tested on the body. A reconfigurable snap-on button antenna module with dedicated circuitry for wearable applications has been illustrated [24]. This button antenna is an off-body antenna, starting with the radiating part which is a monopole fed by a microstrip line for investigating the best structure based on a dielectric resonator [25]. In [26], a novel circularly polarized button antenna is proposed, covering the unlicensed national information infrastructure (U-NII) worldwide band. To date in the discussed button antennas, the antennas have been designed with large dimensions and less gain.

9.If the authors can revise Sections1, 2, and 6 in particular, it is possible to consider that the article will acquire an interesting character for the professional public in the field of antenna technology. Also, I would suggest you to modify the title as well.

R: I have revised all of them, so that the article is complete and logical, and the overall logic will be much smoother. Thank you for your advice. The following secondary title distribution and logical order can support the entire large title.

Line: P 45-145

Here I divide into characteristic analysis: The design and analysis of absorption surface of the button antenna.

Equivalent circuit analysis: The Radiation Direction of Button Antenna is Improved by CRO Metamaterial; Analysis of Gain of the CRW structure of Button Antenna by Quality Factor Q.

Reviewer 2 Report

The authors presented a near-zero-index absorbing metamaterial of button antenna is proposed for pedestrian to vehicular (P2V) communication. The proposed work is presented with good analysis and characterization of a fabricated prototype. However, the reviewer has following concerns:

1. The paper is written in very poor English and style. Some parts are simply difficult to comprehensive. It lacks the overall standard of technical writing in a peer-reviewed journal. Also, some parts of the manuscript indicates lack of mindfulness/carefulness of authors, as example,  line 42: "In this letter....", line 88: "In this chapter....".There are several such mistakes across the manuscript.

2. Could you please elaborately explain how Fig. 6(b) concludes that the rectangular metal ring structure belongs to NZIM?

3. Could you please explain elaborately and properly the relevance of Q-factor in Fig. 8(b)? How and why the rectangular metal ring 1/Q is significantly higher than patch 1/Q? As the reviewer failed to comprehend this statement from the manuscript: "It can be seen from Fig. 8 (b) that the value increment of quality factor Q is much greater than of directivity. It is most suitable to analyze the change of Q value and analyze metamaterials by Q value".

Author Response

The authors presented a near-zero-index absorbing metamaterial of button antenna is proposed for pedestrian to vehicular (P2V) communication. The proposed work is presented with good analysis and characterization of a fabricated prototype. However, the reviewer has following concerns:

  1. The paper is written in very poor English and style. Some parts are simply difficult to comprehensive. It lacks the overall standard of technical writing in a peer-reviewed journal. Also, some parts of the manuscript indicates lack of mindfulness/carefulness of authors, as example, line 42: "In this letter....", line 88: "In this chapter....".There are several such mistakes across the manuscript.

R: All the languages have been revised, and professional organizations have been hired to adjust the grammar and language. Professional English polishing company for English polishing.

  1. Could you please elaborately explain how Fig. 6(b) concludes that the rectangular metal ring structure belongs to NZIM?

R: Here NZIM does not have a specific parameter and technical index, only when the permittivity, magnetic permeability and refraction index tend to 0, become near-zero-index metamaterials. I'm just emphasizing the fact that it has this property, and the material that also has the absorption property. Its specific uses and functions are not described in detail. There is more emphasis on its absorption properties. According to Fresnel Equation, the following formula 1 will not be repeated here, because the subject here is the application of absorbent materials. But this conclusion is not inevitable so it is not mentioned more.

A is absorptivity.

A=1-R=1-|(µr-n)/( µr+n)|2   µr is the permeability, n is refractive index (1)

Line:P62-67

Figure 2 (b) (d) shows the real and imaginary parts of the permittivity ($\varepsilon$), the magnetic permeability($ \mu$) and the refractive index(n) approximately equal to zero by periodic electrical boundary (PEC) and periodic magnetic boundary (PMC) simulation results. The CRO and CRW belong to NZIM.

  1. Could you please explain elaborately and properly the relevance of Q-factor in Fig. 8(b)? How and why the rectangular metal ring 1/Q is significantly higher than patch 1/Q? As the reviewer failed to comprehend this statement from the manuscript: "It can be seen from Fig. 8 (b) that the value increment of quality factor Q is much greater than of directivity. It is most suitable to analyze the change of Q value and analyze metamaterials by Q value".

R: First, this paper emphasizes and rearranges the structure of the paper. Here, metamaterials are not analyzed by mass factor Q. In the fourth paragraph, we emphasize its absorption characteristics, CRO and CRW, and then use this absorbing material as the radiation part of the antenna, and according to the characteristics of the radiation part, such as improving the co-polarized radiation and improving the gain. Then the quality factor Q is used to describe the gain relationship and adjust the real and imaginary parts of the antenna. The method used is equivalent circuit method and basic principle of patch antenna.

Line: P 45-145

Here I divide into characteristic analysis: The design and analysis of absorption surface of the button antenna.

Equivalent circuit analysis: The Radiation Direction of Button Antenna is Improved by CRO Metamaterial; Analysis of Gain of the CRW structure of Button Antenna by Quality Factor Q.

Reviewer 3 Report

The general idea of this work seems interesting and the fabrication and measurement seem to have gone well since the results are consistent. However, almost every other part of the work has serious issues.

1) The authors refer to their structure as a metamaterial absorbing button antenna. The problem is the fact that metamaterials are mainly considered periodic structures which is not the case in this device since one unit cell is utilized.

2) In section 2, the configuration is not adequately described. Specifically, the different colors of copper are confusing. Are they at different layers? Moreover, the four parts mentioned in the manuscript should be highlighted explicitly in Figure 1.

3) In Figure 2, the near-zero behavior is not obvious at all. In particular, the y-axis (epsilon, mu, and n) has very extreme values and the zero or one is not distinguishable.

4) Equations (1) and (2) are not connected with the rest of the text, and there is a mistake in numbering.

5) The authors claim that "the scattering of waves usually causes the scattering of propagating wave on the surface of a patch". How is this clarified? Generally, the description of surface currents in section 3 is not comprehensible enough.

6) It seems that the substrate surface is not covered entirely by metallic parts and a transmitted wave is possible. So, why is the transmittance not accounted for in the absorbance calculation?

7) The layout in sections 4, 5, and 6 is very bad. Specifically, the figures are too far from their mention in the manuscript. Consequently, readability is not facilitated.

8) Why the Q value is more suitable to analyze metamaterials? How is this clarified?

9) How are equations 4-11 evaluated?

10) How is the circuit analysis facilitate the design of the proposed structure? In general, the circuit analysis does not seem relevant since the impedance values can be calculated by the simulation software more accurately. Maybe a completely different work should be conducted to provide the circuit model and compare it with simulations in order to evaluate its accuracy.

11) The language manipulation is very bad and some parts exist that it is very hard to read them.

12) The template maybe is not the correct one.

Author Response

The general idea of this work seems interesting and the fabrication and measurement seem to have gone well since the results are consistent. However, almost every other part of the work has serious issues.

1.The authors refer to their structure as a metamaterial absorbing button antenna. The problem is the fact that metamaterials are mainly considered periodic structures which is not the case in this device since one unit cell is utilized.

R: Cell small absorbers are also valuable because they are portable and scalable to array forms. It is mainly for the future reduction of the overall array size as an application of basic research.

2.In section 2, the configuration is not adequately described. Specifically, the different colors of copper are confusing. Are they at different layers? Moreover, the four parts mentioned in the manuscript should be highlighted explicitly in Figure 1.

R: Because the display is a process drawing, this drawing is drawn by CAD software, so there will be a certain illusion, so it is modified into a simulation drawing, which can better reflect the overall antenna distribution and metal configuration. Thank you for your suggestions and comments.

Line: P 52-53  configure

3.In Figure 2, the near-zero behavior is not obvious at all. In particular, the y-axis (epsilon, mu, and n) has very extreme values and the zero or one is not distinguishable.

R: It has been modified. Due to the large setting parameters and unreasonable setting of frequency range, the figure is fuzzy. Therefore, the setting parameters and frequency range have been clearly set and modified.

Line: P 67-68

4.Equations (1) and (2) are not connected with the rest of the text, and there is a mistake in numbering.

R: Equation 1 and 2 are the absorption characteristics formula, which are explained in detail here because of the simulation model. In my opinion, the model may not be stated, but only the absorption characteristics and absorption Angle of the structure can be stated. It does not emphasize the detailed absorption calculation process, which is too long and is not the innovation point of this article. So let me delete that.

Here NZIM does not have a specific parameter and technical index, only when the permittivity, magnetic permeability and refraction index tend to 0, become near-zero-index metamaterials. I'm just emphasizing the fact that it has this property, and the material that also has the absorption property. Its specific uses and functions are not described in detail. There is more emphasis on its absorption properties. According to Fresnel Equation, the following formula 1 will not be repeated here, because the subject here is the application of absorbent materials. But this conclusion is not inevitable so it is not mentioned more.

A is absorptivity.

A=1-R=1-|(µr-n)/( µr+n)|2   µr is the permeability, n is refractive index (1)

5.The authors claim that "the scattering of waves usually causes the scattering of propagating wave on the surface of a patch". How is this clarified? Generally, the description of surface currents in section 3 is not comprehensible enough.

R: I think the more understandable explanation of the correlation between NZIM characteristics and absorption characteristics is not complete. First, NZIM is a principle of physical light lens used for light concentration, so that both refractive index and projection are concentrated at one point. And absorption is the absorption, the elimination, the conversion of energy in some way into other properties like temperature, heat and so on. We're just describing the absorption properties, because the principle relates to the physics and the actual properties of the structure. This explanation is immature. This phenomenon will not be further elaborated here.

  6. It seems that the substrate surface is not covered entirely by metallic parts and a transmitted wave is possible. So, why is the transmittance not accounted for in the absorbance calculation?

 R: A(w)=1-|S11(w)|2-|S21(w)|2   |S11(w)|2 is reflected power |S21(w)|2 is transmitted power. A(w) is absorbance. It's better to say that minus projection and reflection is the absorption rate, which is by far the dominant explanation. According to Fresnel Equation, the following formula 2 will not be repeated here, because the subject here is the application of absorbent materials. But this conclusion is not inevitable so it is not mentioned more.

A is absorptivity.

A=1-R=1-|(µr-n)/( µr+n)|2   µr is the permeability, n is refractive index (2)

   7. The layout in sections 4, 5, and 6 is very bad. Specifically, the figures are too far from their mention in the manuscript. Consequently, readability is not facilitated.

R: Part 4,5 and 6 are modified in detail, Part 4 describes its absorption characteristics, and Part 5 applies it to button antenna. CRO can improve the co-polarized radiation and make the antenna become directional antenna from omnidirectional antenna. The sixth part is to embed microstrip line on the basis of CRO to make it become CRW, and make equivalent circuit analysis for gain increase, mainly using quality factor Q analysis.

Parts 4,5, and 6 have been revised:line P 83-213

   8. Why the Q value is more suitable to analyze metamaterials? How is this clarified?

R: Since Q is related to gain G and bandwidth BW, the gain analysis must analyze Q value and its related impedance information. Firstly, metamaterial application of button antenna is explained instead of equivalent circuit. The application of CRO (copper coil) and CRW (copper coil embedded with microstrip wire) in antenna is analyzed by mass factor Q. Metamaterial Properties Part 3 has been revised and detailed.

Line:P 120-141  Analysis of Gain of the CRW structure of Button Antenna by Quality Factor Q.

First, analyzes the improvement of the gain of the button antenna by quality factor Q is analyzed. Second, the equivalent circuit method is used to analyze the embedded microstrip lines. Finally, the effect of its quality factor and CRW structure on the gain of the button antenna is analyzed.As seen in the previous section, the largest change in the CRO structure is the quality factor Q, as shown in Equation 11. In this way, the only parameter that affects antenna gain is the value Q. Therefore, it becomes an inevitable scheme to further improve the quality factor by embedding microstrip lines to improve the width of the copper coil of the CRO. The next step is to analyze the relationship between the $\rm T_1$ length and quality factor Q, and further analyze the change of the Q value after adding other embedded microstrip lines. According to Equation 3, Q is related to the real and imaginary parts of the impedance. Therefore, this chapter mainly analyzes the impedance after adding the $\rm T_1$ symmetrical microstrip line.

   9. How are equations 4-11 evaluated?

R: I have made comprehensive modifications. Here I mainly describe the relationship between the coupling distance of CRO and the microstrip line width and the mass factor Q, so as to find out which parameter has the greatest influence on Q, and further adopt specific methods to improve Q and further improve antenna gain. As shown in equation 11. This is also reflected by adding equation 10.

Formula 10 further illustrates the relationship and correlation. P127-128

  10. How is the circuit analysis facilitate the design of the proposed structure? In general, the circuit analysis does not seem relevant since the impedance values can be calculated by the simulation software more accurately. Maybe a completely different work should be conducted to provide the circuit model and compare it with simulations in order to evaluate its accuracy.

R: Here, the formula is combined with simulation, mainly the influence factor affects the Q value of the mass factor and the D value of the direction coefficient, and the real and imaginary parts of the impedance are derived according to the mass factor, and then the equivalent circuit method is used as the judgment basis. For example, in Formula 10, we can determine that the mass factor is proportional to the sum of the microstrip line widths and inversely proportional to the sum of the coupling distance. It is more reasonable to increase the width of microstrip line than to reduce the coupling distance, and it is proposed to embed microstrip line, which has a certain rationality.

Equations 10 and 11 illustrate the problem: P 127-128 ;P 141-142

11. The language manipulation is very bad and some parts exist that it is very hard to read them.

R: It has been completely revised and polished in English by Professional English polishing company.

12. The template maybe is not the correct one.

R: I have consulted the editor, and this template will be used for the review template, and the prescribed template will be used after the review. Thanks for reminding me.

Round 2

Reviewer 1 Report

The authors have improved the manuscript from the previous version. However, Still, some more improvements are required.

The conclusion section is not appropriate

My earlier comments, such as comment1,3, and 9  are not properly justified

In response to comment 3, the FR-4 loss tangent was mentioned as 0.0027. Please verify it once again

Author Response

The conclusion section is not appropriate

I've already made some changes here:

Line P210-217

An absorbing material on a button antenna for P2V communication is proposed, fabricated, and measured, which provide safety service to road workers on the highway or in a road environment. The maximum distance of measurement between the absorbing material of the button antenna and the test antenna is less than 150 m, which is a safe distance for traffic. This antenna can achieve a maximum gain of 5.5 dBi and absorption of 92\% at 7.6 GHz. Comparing with other similar antennas in the Table 2, the proposed antenna is very compact in size and achieves an excellent gain and reasonable efficiency, which the bandwidth is only 4\%. The modified absorbing material structure and the detail considerations for wearable scenarios make it a very suitable candidate for on-body applications.

My earlier comments, such as comment1, 3, and 9 are not properly justified

1)It is decisive for me that you have relevant results in the given research. I want to draw your attention to the fact that the abstract in which you demonstrate the meaningfulness of scientific conclusions is very focused on determining your experiment and not clarifying your improvements and what is the goal of your manuscript Pedestrian to Vehicle Communication. This generally discourages the reader from continuing to read it.

R: First, this button antenna is used in P2V communication in the human body.

Second, it is characterized by miniaturization and portability.

  Finally, it is necessary to compare the characteristics and advantages of the same type antenna.

Therefore, the experimental results and simulation process are extremely important. Here, P2V is only the application scenario, and the effective safe distance of its antenna is determined. The abstract and introduction are detailed here, and the final summary also highlights the characteristics of the antenna.

Abstract modification: line P 1-9: In this paper, an absorbing material on a button antenna is proposed for pedestrian-to-vehicle (P2V) communication, providing safety service to road workers on the highway or in a road environment. The button antenna is small in size and easy to carry for carriers.

Introduction modification: line P 15-21: To provide a good communication service to the P2V communication system, one should ensure that a minimum broadcast distance is achieved such that an adequate road safety level can be provided[9]-[15]. Therefore, directive antennas have been proposed to improve connectivity and security for vehicular communication and autonomous systems [16]-[18]. However, making the antenna smaller makes it easier to wear or carry. The button antenna is the most comfortable technology because it can be easily integrated into human clothes and provides stable performance due to its rigidity [19] [20].

2)Does the FR-4 material have loss? Could you provide the radiation efficiency of your antenna at the operating frequency?

 R: I have modified it to 0.027, thank you for your careful reference

  Configuration : line P51

3)If the authors can revise Sections1, 2, and 6 in particular, it is possible to consider that the article will acquire an interesting character for the professional public in the field of antenna technology. Also, I would suggest you to modify the title as well.

R: Part 1, 2, and 6 are the main process of antenna type articles, and also the main line of some articles. Here is the single column, which means that the layout pays more attention to the logic of the article, so the basic process of antenna article is used for typesetting. The title has been changed to “Absorbing material of Button Antenna with Directional Radiation of High Gain for P2V Communication”

    Title modification:

Absorbing material of Button Antenna with Directional Radiation of High Gain for P2V Communication

In response to comment 3, the FR-4 loss tangent was mentioned as 0.0027. Please verify it once again

R: I have modified it to 0.027, thank you for your careful reference.

Configuration : line P51

Reviewer 2 Report

I have no further comments. I recommend to accept the manuscript for publication. 

Author Response

I have no further comments. I recommend to accept the manuscript for publication.

R: Thanks for you encouraging suggestion.

Reviewer 3 Report

The authors conducted some changes in the manuscript and it has been improved. But, there are still many points to clarify:

1) Indeed, unit-cell resonators can be used to acquire various interesting features. However, they are not metamaterials and the authors have to change it both in the title and in the manuscript since it is misleading.

2) How are the electric permittivity, electric permeability, and refractive index have been calculated in Figure 2? A proper reference is required.

3) The authors removed the part with the extraction of absorptivity, but there are various results that rely on it. Moreover, the absorptivity is not straightforwardly extracted by simulation software. Thus, the calculation formula must be mentioned to clarify the results. Of course, both the reflection and transmission coefficients should be accounted for since there is no total reflection.

4) Table 2 has some weird results. In particular, the upper figure of the right side shows that the case of adding P1 and T1 retains the 1/Q value below 5 from 3 to 9 GHz. However, in the middle figure (again right side) the P1 and T1 curve is lower than 5 only from 4 to 6. Please explain this.

5) The reference to Table 2 is not correct (it is noted as Table 1 in the text). Moreover, it should be transformed into a figure.

Author Response

1)Indeed, unit-cell resonators can be used to acquire various interesting features. However, they are not metamaterials and the authors have to change it both in the title and in the manuscript since it is misleading.

R: The title has been changed to “Absorbing material of Button Antenna with Directional Radiation of High Gain for P2V Communication”. Originally NZIM was a metamaterial, but due to concerns about absorptive properties, the topic of absorptive materials was adopted after consideration of repeated expert opinions.

Absorbing material of Button Antenna with Directional Radiation of High Gain for P2V Communication

2)How are the electric permittivity, electric permeability, and refractive index have been calculated in Figure 2? A proper reference is required.

R: I've added to the equation, equations 1 and 2.

 According to the literature: eshmukh R , Marathe D , Kulat K D ." Microstrip Patch Antenna Gain Enhancement using Near-zero Index Metamaterial Superstrate (NZIM Lens)" , 2019 10th International Conference on Computing, Communication and Networking Technologies, 2019.

Line P58-59

3)The authors removed the part with the extraction of absorptivity, but there are various results that rely on it. Moreover, the absorptivity is not straightforwardly extracted by simulation software. Thus, the calculation formula must be mentioned to clarify the results. Of course, both the reflection and transmission coefficients should be accounted for since there is no total reflection.

R: I've added to the equation, equations 3

According to the literature: Garg P , Jain P ." Isolation Improvement of MIMO Antenna Using a Novel Flower Shaped Metamaterial Absorber at 5.5 GHz WiMAX Band". IEEE Transactions on Circuits and Systems II: Express Briefs, 4,2020, PP. 675-679.

Line P74-75

4)Table 2 has some weird results. In particular, the upper figure of the right side shows that the case of adding P1 and T1 retains the 1/Q value below 5 from 3 to 9 GHz. However, in the middle figure (again right side) the P1 and T1 curve is lower than 5 only from 4 to 6. Please explain this.

R: Here, three embedded microstrip lines are adopted on the left. Figure (a) shows the change of Q value after P1 is added for the first time. By analogy, we need to find the minimum value of 1/Q, that is, the maximum value of Q, to verify the maximum value of radiation in Figure (c). Therefore, (a) is a comparison graph, and (b) is to find the minimum value of 1/Q after the addition of three microstrip lines. Since the curve is required to be simple and clear, this is a part of the graph to compare the three 1/Q clearest graph.

Line 160-176

As shown in Figure (a) and (b) of Figure 16, with the addition of microstrip lines T1, T2, T3, P1 P2 and P3, The value of 1/Q is getting smaller and smaller, so we have to optimize for the smallest 1/Q, which is the maximum value of Q. The HFSS software is adopted for optimization, and the minimum value of 1/Q at 7.6GHz in Figure (b). Figure (c) verifies that the maximum value of Q is the maximum value of antenna gain by gain of co-planar polarization.

5)The reference to Table 2 is not correct (it is noted as Table 1 in the text). Moreover, it should be transformed into a figure.

R: Has been changed to Figure 16.

Line 176-177

CRW analysis: adding  P1 current distribution (case 1); adding  P2 and  T2 current distribution (case 2); adding P3  and  T3 current distribution (case 3); Comparison diagram of 1/Q after adding P1 (a); the minimum value of 1/Q after CRW optimization (b) ; maximum gain after CRW optimization (c)